# Antifungal Activities of Natural Products and Their Hybrid Molecules

**DOI:** 10.3390/pharmaceutics15122673

**Published:** 2023-11-25

**Authors:** Vuyolwethu Khwaza, Blessing A. Aderibigbe

**Affiliations:** Department of Chemistry, University of Fort Hare, Alice Campus, Alice 5700, Eastern Cape, South Africa

**Keywords:** natural product, hybrid molecules, fungal infections, antifungal drugs, drug resistance

## Abstract

The increasing cases of drug resistance and high toxicity associated with the currently used antifungal agents are a worldwide public health concern. There is an urgent need to develop new antifungal drugs with unique target mechanisms. Plant-based compounds, such as carvacrol, eugenol, coumarin, cinnamaldehyde, curcumin, thymol, etc., have been explored for the development of promising antifungal agents due to their diverse biological activities, lack of toxicity, and availability. However, researchers around the world are unable to fully utilize the potential of natural products due to limitations, such as their poor bioavailability and aqueous solubility. The development of hybrid molecules containing natural products is a promising synthetic approach to overcome these limitations and control microbes’ capability to develop resistance. Based on the potential advantages of hybrid compounds containing natural products to improve antifungal activity, there have been different reported synthesized hybrid compounds. This paper reviews different literature to report the potential antifungal activities of hybrid compounds containing natural products.

## 1. Introduction

Fungal infections pose a significant concern to public health and can result in life-threatening acute diseases (e.g., *cryptococcosis* and invasive *aspergillosis*), severe chronic diseases, such as allergic bronchopulmonary *aspergillosis*, etc., or can present less-threatening superficial infections (e.g., Candida vaginitis or oral candidiasis) [1]. Fungal species from the four genera of *Candida*, *Pneumocystis, Cryptococcus*, and *Pneumocystis* result in more than 90% of recorded fungal-associated deaths [2]. An approximate estimate of 3 million cases of chronic pulmonary *aspergillosis* have been reported globally [3], including 700,000 cases of invasive candidiasis [4], 500,000 cases of Pneumocystis *jirovecii pneumonia* [1], 250,000 cases of invasive *aspergillosis* [1], 220,000 cases of AIDS-associated *cryptococcosis* [5], and 100,000 cases of disseminated histoplasmosis [6]. According to the World Health Organization (WHO), the fungal strains are becoming more widespread and resistant to the currently used antifungal agents. Four classes of antifungal drugs are available with a few of them undergoing clinical trials [7]. Fungal diseases are the cause of a higher global mortality rate than breast cancer or malaria [8]. Recent research indicates that the fourth most common cause of nosocomial bloodstream infections is systemic infections, resulting from the Candida species. Over 90% of invasive infections are caused by opportunistic fungal pathogens [9,10,11]. There are at least 15 different species in this family, but only five of them have the potential to cause invasive infections, causing a high rate of mortality. Among the *Candida* species that cause invasive infections are *Candida tropicalis, Candida krusei, Candida albicans*, *Candida glabrata*, and *Candida parapsilosis* [12,13,14]. *Candida albicans* can lead to fatal invasive infections of the mucosa in immunocompromised people who are receiving chemotherapy for cancer [9,10,11]. *Candida auris*, a newly discovered fungal strain, is an emerging multidrug-resistant microorganism that poses a serious threat to human health worldwide [15]. Although *C. auris* is the newest species of *Candida*, it is resistant to all the available antifungal drugs [16]. Furthermore, invasive fungal infections have worsened the clinical course of COVID-19 and are associated with an increased mortality rate, particularly in patients admitted to an intensive care unit (ICU) [17]. One of the best methods for minimizing fungal infections is the use of antifungal medications. However, the widespread administration of antifungal medications has increased microbial resistance, which poses a global health concern [18]. For a treatment to be effective, targeted delivery of the administered therapeutic to the site of infection at a concentration high enough to induce antimicrobial activity is required. Many therapeutic drugs’ pharmacokinetics are known, however, there is still a lack of knowledge regarding the drug uptake at the infection sites [1]. As a result, certain microbes are exposed to drugs at suboptimal levels, resulting in their continuous existence even after appropriate treatments and may develop into subclinical reservoirs that spread infections. The above-mentioned factors influence microbial resistance and a selection of pathogens can multiply even after being exposed to therapeutic concentrations of antifungals. These pathogens are a major cause of drug resistance during treatment. Microbial resistance encompasses two types of strains: primary resistant strains, which are naturally less sensitive to a particular antifungal agent, and secondary resistant strains, which develop resistance characteristics after being exposed to a drug and become susceptible strains. Primary resistant microbial strains frequently exhibit different degrees of expression for the molecular mechanism underlying acquired resistance. Azole resistance emerges through multiple mechanisms such as drug target overexpression or alteration, drug transporter upregulation, or cellular modifications that mitigate drug toxicity or facilitate tolerance to drug-induced stress. Polyene’s resistance generally involves depletion of the target ergosterol attributable to loss-of-function mutations in ergosterol biosynthetic genes [19]. To address these problems, it is essential to develop new antifungal agents with novel mechanisms and high selectivity. Therefore, research into the potential use of plant-based antimicrobial compounds for the treatment of fungal diseases has attracted a lot of attention [20]. Research on antifungal drugs is driven by the pressing need to discover effective treatment options that are safe for humans and beneficial to the environment. Plant-based antifungal drugs undoubtedly offer such options, not only because of their tremendous structural varieties but also because of their broad range of pharmacological activities. The majority of commercially available antifungals are made from natural products [21]. Most natural products have been discovered to be superior to synthetic drugs due to their selectivity in binding to fungi and their non-toxic nature. Their underlying molecular targets or mode of action are listed in Table 1 and they include fungal cell membranes, cell walls, and numerous organelles, in addition to acting as preventive agents [22]. Innovative drugs developed for the treatment of various infectious diseases, including candidiasis, etc., have been developed by modifying the structures of natural products to increase their effectiveness, solubility, and safety [23,24].

The molecular hybridization of natural products is a novel idea in drug discovery and development, and it plays a crucial role in the development of new molecules with enhanced biological activities. To overcome the problem of drug resistance against a variety of microbial pathogens, the idea of hybridization can be explored to obtain more effective therapeutic candidates against microbial pathogens [25]. The structural combination of two or more natural product fragments results in novel structures with better biological activity than the parent molecules [26,27]. It is a useful strategy for boosting bioactive compounds’ inhibitory activity, improving pharmacodynamic and pharmacokinetic properties, and minimizing toxicity. Previous research studies have demonstrated that hybrid compounds containing natural product scaffolds displayed enhanced therapeutic outcomes [28,29,30,31]. This short review discusses the potential antifungal activities of selected natural products and their hybrid molecules, as well as their structure–activity relationships (SARs), and provides recommendations on a new direction for the design and development of natural product-based hybrid molecules with high efficacy against drug-resistant fungal pathogens.

**Table 1 pharmaceutics-15-02673-t001:** Antifungal mode of action of some natural products.

Natural Products	Mode of Antifungal Activity	Bibliography
Carvacrol	Alters the membrane structure of the fungal cell in *C. albicans.* Disrupts ergosterol biosynthesis and membrane integrity against *Candida species*. Reduces spore germination, mycelia growth, aflatoxin production, and the ergosterol content of *A. flavus.* Binds to exogenous ergosterol, promoting interaction with cholesterol in *Cryptococcus neoformans*. Modulates the expression and activity of antioxidant enzymes in *Candida auris.*	[32,33,34,35,36]
Eugenol	Exerts antifungal effects on the cell wall and cell membrane of *Trichophyton rubrum by* inhibiting ergosterol biosynthesis.	[37]
Coumarin	Induces a series of apoptotic features in *C. albicans*, including phosphatidylserine externalization, DNA fragmentation, and nuclear condensation.	[14]
Cinnamaldehyde	Inhibited germ tube formation, adhesion to epithelial cells, and hydrolytic enzyme secretion of *C. albicans.* Affects the ergosterol biosynthetic processes and leads to the disruption of the cell membrane integrity of *Fusarium sambucinum.* Exhibit its antifungal activity against *G. citri-aurantii* by interfering with the formation of cell walls, resulting in the damage of cell wall permeability and integrity	[38,39]
Curcumin	Exerts antifungal activity via disrupting fungal plasma membrane in *C. albicans.* Alters the membrane-associated properties of ATPase activity, ergosterol biosynthesis, and proteinase secretion in *C. albicans* and *Candida glabrata*. Exerts fungal cell membrane disruption and the inhibition of ergosterol synthesis, respiration, succinate dehydrogenase, and NADH oxidase. Induces reactive oxygen species and triggers early apoptosis but prevents hyphae development by targeting the global repressor TUP1 in *Candida albicans*	[40,41,42,43]
Thymol	Affects intracellular calcium homeostasis by suppressing the expression of genes involved in the calcium transporters. Decreases expression levels of genes required for *N*-glycosylation, thereby reducing protein glycosylation. Decreases ergosterol contents in a HOG pathway-dependent manner in *Cryptococcus neoformans.* Induces Lipid Peroxidation and disrupts Ergosterol Biosynthesis in *Fusarium graminearum*. Produces reactive oxygen species (ROS) accumulation and destroys the integrity of the cell wall and cell membrane by inhibiting the genes involved in the cell wall and cell membrane synthesis.	[44,45,46]

## 2. Natural Products

### 2.1. Coumarin

Coumarin (**1**, Figure 1) is among the intriguing class of natural products that exist as bio-active heterocycles with distinct pharmacological and physical properties. Coumarins are abundant in plants and can also be extracted from bacteria and fungi. There are around 1300 coumarins isolated from natural sources, and their medicinal uses depend on the specific substitution patterns [47]. According to the research findings by Xu et al., coumarin has antibiofilm activity against *C. albicans*, which entails reducing adherence and morphological transition. Some research reports revealed that coumarin blocks the transition from yeast to hyphae, possibly by way of the cAMP pathway [48]. Among all the pharmacological activities of coumarins, it was discovered that coumarins can inhibit fungal development, depending on the substituents attached to the coumarin core. As a result, various coumarin-based hybrid molecules have been studied as potentially effective drugs in the prevention and control of fungal infections. The incorporation of a coumarin scaffold with other bioactive molecules can result in potent hybrid compounds that exhibit enhanced therapeutic properties [49]. It is possible that combining the coumarin moiety with other antimicrobial agents is a potential approach for developing new therapeutics. Modification of coumarins is of great interest due to their distinctive structural characteristics and pharmacological properties, such as antiviral [50,51], anti-inflammatory [52,53], antibacterial [54], anticancer [55,56], and antioxidant [57] activities. Several synthesized molecules containing coumarin rings have been reported to be effective against multi-drug resistant (MDR) microorganisms [58,59]. Furthermore, some coumarin-based hybrid molecules, including novobiocin, clorobiocin, and coumermycin A1, have already been used in clinical practice to treat a variety of bacterial infections [58,59]. Numerous studies have reported the synthesis of new hybrid compounds containing coumarin pharmacophore(s) with improved antifungal properties [60,61].

Zhang et al. prepared a variety of coumarin-based hybrid compounds with a pyrrole pharmacophore and investigated their antifungal activity (in vitro) against six phytopathogenic fungi. Among the synthesized hybrid molecules that exhibited potential fungicidal activities against the tested fungi, compounds **2**, **3**, **4**, **5**, and **6** (in Figure 2) exhibited significant antifungal effects against *Rhizoctorzia solani* with EC_50_ values of 3.94, 6.25, 6.38, 7.67, and 7.75 µg/mL, respectively. The aforementioned activities were more potent than the commercially available fungicides, Osthole (9.79 µg/mL) and Boscalid (11.52 µg/mL). According to their structure–activity relationship, hybrid molecules with a hydroxyl group (OH) at R_2_, a methyl group (CH_3_) at R_3_, and an H atom at R_4_ position, such as hybrids **2** and **6**, inhibited *Botrytis cinerea, Gibberella zeae*, and *Rhizoctorzia solani* better than those with CH_3_ at R_2_ with H at R_3_ snd R_4_ positions [62]. Sadgir et al. synthesized and characterized a series of coumarin with attached thiazole derivatives to determine their antibacterial and antifungal activities. The antibacterial activity of the synthesized coumarin-thiazole hybrids was tested against *S. pyogenes*, *S. aureus*, *P. aeruginosa*, and *E. coli* strains, while the antifungal activity was tested against *A. clavatus*, *A. niger*, and *C. albicans* strains. When compared to the standard drug, some of the synthesized compounds demonstrated good antibacterial efficacy against strains of *P. aeruginosa* and *E. coli* but had weak antifungal activity. The ADME profile of these molecules demonstrated favorable pharmacological features [63]. Trivedi et al. employed a click chemistry synthetic approach to prepare novel coumarin-based 1,2,3-triazole hybrid molecules from 4-hydroxy coumarin with several substituted azides. The synthesised triazole hybrids were evaluated for their antifungal efficacy against various fungal species, including *Penicillium* spp. *Ganoderma* spp., *A. flavus*, and *A. niger* using agar plate method. The antifungal activity of the three new triazole hybrid molecules **7**, **8**, and **9** (in Figure 2) was either moderate or good [64]. Yang et al. prepared coumarin thiazoles with a trifluoromethyl group via a solvent-free one-pot reaction in which 3-(trifluoroacetyl) coumarin was used as a precursor. Compound **10** in Figure 2 demonstrated the highest inhibitor rates of 89% and 93.4% at a concentration of 0.5 mg/mL against *F. graminearum* and *C. lunata*, respectively, while compound **11** demonstrated the highest inhibitor rate of 74% at a concentration of 0.5 mg/mL against *F. moniliforme* among the synthesized compounds. It was shown that the presence of a substituent in the 3-position of the thiazole ring or the naphthalene ring in coumarin promoted antifungal properties that are unfavorable for the suppression of the fungal strain *F. graminearum*. Comparing compound **10** to compound **12**, the presence of a trifluoromethyl group significantly increased the coumarin antifungal activity [65]. Al-Amiery et al. developed two potent coumarin hybrid molecules 4-((5-mercapto-4-phenyl-4H-1,2,4-triazol-3-yl)-methoxy)-2H-chromen-2-one and 4-((5-(phenylamino)-1,3,4-thiadiazol-2-yl)-methoxy)-2H-chromen-2-one against two fungal species such as *Aspergillus niger* and *Candida albicans.* When compared to the standard antifungal drug fluconazole, these two compounds **13** and **14** displayed good antifungal activity. According to the SARS of these coumarin hybrid molecules, the presence of the amino derivative substituents in the coumarin pharmacophore was significant to their pharmacological activity [66].

### 2.2. Essential Oils

Essential oils are volatile organic compounds that are produced by aromatic plants. These compounds include terpenes, terpenoids, and aromatic or aliphatic molecules. Recently, their numerous benefits have been explored significantly in the sanitary, food, perfume, cosmetic, and pharmaceutical industries. Several studies have revealed the unique features of essential oils, such as their chemical structures, mode of action, etc. They contain organic compounds that can be divided into four classes based on their chemical structures: terpenes (mono and sesquiterpene), terpenoids (phenols, alcohols, ketones, ethers, esters, aldehydes, and epoxides), phenylpropenes, and aromatics containing sulfur and nitrogen [67]. These essential oils are produced from different plant species through various biosynthetic pathways for defense mechanisms against varieties of biotic or abiotic factors. The importance of essential oils goes beyond just preserving plants but is a mine of potential therapeutics with pharmacological properties that need to be explored [68]. Numerous studies have shown that essential oils exhibit insecticidal, anticancer, anti-inflammatory, antimicrobial, and antifungal activities [69,70,71,72]. However, there is a tremendous need for the large-scale manufacturing of essential oils and the improvement of their therapeutic value in the food and pharmaceutical industries [73]. Essential oils are promising effective substitutes or additions to synthetic molecules without the capability to induce unwanted adverse effects. New molecules developed from essential oils have been able to overcome the pharmacological drawbacks associated with essential oils, such as poor water solubility and high volatility, while enhancing their efficacy against fungal pathogens.

#### 2.2.1. Terpenoids

Terpenoids, also referred to as isoprenoids or terpenes, make up the biggest group of natural compounds with over 30,000 distinct structures [22]. The antifungal activity of some terpenoids against human and plant pathogenic fungi has been documented. Most of these terpenoids are of plant origin. Many studies have demonstrated the ability of terpenoids to kill a wide range of harmful fungal and bacterial pathogens such as *Candida albicans*, *Staphylococcus aureus*, and *Pseudomonas aeruginosa*, including their drug-resistant strains. Terpenoids are the main active compounds in essential oils and some of them exhibit strong antifungal effects by triggering mitochondrial dysfunction and disrupting cell membrane integrity in *Saccharomyces* [74,75], inhibiting *C. albicans* growth, arresting the cell cycle [76], and inhibiting morphogenesis, adhesion, and biofilm formation by *C. albicans* [77]. Among the bioactive terpenoids with antifungal effects, carvacrol (**15**), thymol (**16**) and eugenol (**17**) (shown in Figure 3) have received significant research attention. 

#### 2.2.2. Carvacrol and Thymol

The antibacterial and antifungal properties of carvacrol have been thoroughly studied. Carvacrol (**15**, in Figure 3) exhibits antifungal properties against a variety of *Candida* species such as *C. albicans, C. glabrata*, and *C. parapsilosis* [32]. It is also effective against fungal strains which include *Penicillium rubrum*, *Alternaria alternate, Aspergillus niger*, *Trichoderma viride*, and *Aspergillus favus* [78]. Carvacrol can combat some microbial pathogens, resulting from the presence of a phenolic hydroxyl group [79]. The antibacterial properties of Carvacrol are attributed to its significant impact on the cytoplasmatic membrane’s structural and functional properties, involving its disruption and interaction with the membrane proteins and intracellular targets [80]. Recently, Niu et al. reported carvacrol’s effect on mitochondrial malfunction, ROS generation, membrane rupture, and apoptosis in *Candida albicans* [81]. Carvacrol also breaks down the fungal envelope, preventing the formation of ergosterol and interfering with the membrane integrity. It alters the capacity of proteins to fold, causing endoplasmic reticulum stress in *C. albicans* [78]. Hybrid compounds containing carvacrol and other bioactive molecules have been synthesized and have been found to exhibit a wide range of biological activities. The combination of carvacrol with known antimicrobial drugs offers effective alternative drug therapies for treating fungal infections, such as *cutaneous pythiosis*, etc., with promising synergistic effects [82].

Pete et al. developed several carvacrol-based hybrid compounds, **18**–**20** by structurally fusing carvacrol with the benzoylphenyl urea linkage (Figure 4). The antifungal activity of hybrids **18**, **19**, and **20** was superior to carvacrol and lufenuron against human pathogens, such as *Cryptococcus neoformans* and *Candida albicans.* Compounds **18** and **20** revealed the presence of a chloride group in the para position enhanced their antifungal activities [83]. Wang et al. synthesized twenty ester-linked hybrid derivatives with various heterocyclic units which were more efficient in inhibiting the fungal pathogens, and further investigated the effects of adding various heterocyclic units to thymol and carvacrol esters. Their results revealed a range of carvacrol and thymol esters with good to exceptional antifungal properties. Compounds **21**, **22**, and **23** were the most potent antifungal compounds against *R. solani*, with equivalent or superior antifungal properties compared to their parent molecules and chlorothalonil. Preliminary research showed that the addition of pyridine, thiophene, and furan units increased the antifungal activities of thymol and carvacrol esters on *Botrytis cinerea*. Incorporating a bromine atom on the para position of the benzene molecule also enhanced the antifungal activity of the compounds [84]. Bagul et al. modified the active groups on the carvacrol moiety with a hydrazide-based sulfonamide to develop potent antimicrobial agents. The newly developed hybrid molecules were tested for their antibacterial properties against three bacterial pathogens (*Escherichia coli, Staphylococcus aureus*, and *Bacillus subtilis*) and three fungal strains (*A. fumigatus, A. flavus,* and *Aspergillus niger*). Compounds **24** and **25** in Figure 4 exhibited good antifungal activity against three selected fungal strains, with *Aspergillus fumigatus* being the most sensitive [85].

Thymol (**16**, in Figure 3), an isomer of carvacrol and a phenol derivative of terpenoids, is extracted from the essential oils of various Lamiaceae plant species, including those from the genera *Thymus, Monarda*, *Thymbra, Saturej, Origanum,* and *Ocimu* [86,87,88]. Thymol, carvacrol, and eugenol were investigated by Doke et al. for their antifungal effects when combined with fluconazole, a well-known antifungal drug, to combat mature biofilms, the development of biofilms, and planktonic cells made by *C. albicans*. The combination of thymol and fluconazole did not induce interaction with the planktonic cells, however, carvacrol and eugenol were found to induce synergistic antifungal effects when coupled with fluconazole [89]. de Vasconcelos et al. compared the antifungal activity of thymol to that of miconazole to assess the cell viability of *C. albicans* biofilms. Thymol and miconazole were used to treat biofilms that had formed on the surface of acrylic resin discs used as dental prosthetics. Thymol and miconazole were effective as antifungal agents by lowering the cell viability of *C. albicans* biofilm growth when used in fluorescence imaging [90]. Another study, conducted by Shu et al., evaluated the impact of thymol on the growth of *C. albicans* biofilms. Thymol inhibited *C. albicans* growth and biofilm formation in a dose-dependent manner at a concentration ranging from 64 µg/mL to 128 µg/mL. The p38 MAPK signaling pathway played a crucial role in the underlying mechanism of thymol activity. Additionally, via the p38 MAPK signaling pathway, thymol exhibits protective effects against *C. albicans* infection and also maintains the innate immune system [91]. Thymol’s antifungal effectiveness against *C. albicans*, *C. krusei*, and *C. tropicalis* strains was examined by Castro et al. to investigate its mode of action and synergistic effect when combined with a synthetic antifungal drug, nystatin. Thymol demonstrated antifungal activity with a MIC value of 78 µg/mL against *C. tropicalis* and 39 µg/mL against *C. krusei* and *C. albicans*. The antifungal activity was unaffected by the presence of sorbitol, however, the presence of exogenous ergosterol caused thymol’s MIC value against *C. albicans* to rise by eight times, from 39.0 to 312.5 µg/mL. The combination of thymol and nystatin reduced the MIC values by 87.4%, yielding a 0.25 FIC index. [92]. The role that thymol plays in medicinal chemistry has inspired numerous researchers to explore its wide range of biological activities. Numerous thymol derivatives have been developed and evaluated for their biological characteristics; we can specifically mention glucosides [93], Mannich bases [94], ethers [95], aldehydes [96], esters [97], azo dyes [98], and formylation products [99]. According to Desai et al., thymol moiety linked to pyrazole, isoxazole, and pyridine motifs produced compounds with promising antimicrobial activities [100].

#### 2.2.3. Eugenol

Eugenol (**17**, in Figure 3) is one of the phenolic monoterpene molecules from the phenylpropanoid family with strong antimicrobial activities [101]. Its derivatives have shown improved antimicrobial activities and non-toxic properties [102]. Carrasco et al. reported nitro and acylated derivatives of eugenol with antifungal activity that was comparable to that of STD medications. The eugenol derivative 4-allyl-2-OMe5-NO_2_-phenol had the highest MIC value against *Cryptococcus neoformans, dermatophytes,* and *Candida albicans* strains. According to SARs, the significant antifungal activity is attributed to the allyl substituent at C-4, a hydroxyl group at C-1, a methoxy group at C-2, and the presence of the nitro groups on the aromatic ring [103]. De Carvalho et al. synthesized benzoxazole-type hybrid molecules of eugenol (**26**, **27**, **28**, and **29**, in Figure 5). They were five times more effective than eugenol against *C. glabrata* and *C. albicans*. Compounds **27** and **29** were effective against the fluconazole-resistant strains of *C. krusei* [104]. The anti-*Candida* potential of eugenol hybrid molecules prepared by phenol group modification was reported by Dutra et al., together with the interaction modes at the lanosterol-14-demethylase site, the morphological changes, and the metabolism-mediated cytotoxicity. Compounds **30** and **31** in Figure 5 were the most effective compounds against *Candida albicans* and *C. parapsilosis*, with MIC values ranging between 50 and 100 µg/mL. SEM analysis of compounds **30** and **31** showed changes in *C. albicans* and *C. parapsilosis* envelope architectures, similar to the morphological changes caused by eugenol and fluconazole. Docking results showed similar binding patterns of compound **30** with a cytotoxicity profile as fluconazole and posaconazole, suggesting that they are potential anti*-Candida* agents [105]. Eugenol-based hybrid molecules **32** in Figure 5 showed notable antifungal activity against *C. auris*, resulting from its capability to induce apoptosis and cell cycle arrest. This compound’s low toxicity towards red blood cells compared to eugenol reveals its safe usage in vivo [106]. P’eret et al. demonstrated the antifungal activity of novel imidazoles and 1,2,4-triazoles hybrid molecules synthesized from dihydroeugenol and eugenol. The imidazole-based hybrid molecules **33**, **34**, **35**, and **36** in Figure 5 exhibited potent antifungal activity against *Candida sp.* and *Cryptococcus gattii*, with MIC values ranging from 4.6 to 75.3 μM. The most effective azole against *Candida albicans* with (MIC: 4.6 µM) was eugenol-imidazole **35**, which was 32 times more effective than miconazole (MIC: 150.2 µM) with no cytotoxic effect, and a selectivity index >28. Notably, the imidazole hybrid **36** (MIC: 36.4 μM) fused with dihydro-eugenol was over 5 times more effective than fluconazole (MIC: 209.0 µM) and two times more potent than miconazole (MIC: 74.9 μM) against multi-resistant *Candida auris.* By the behavior seen with the control drugs, miconazole and fluconazole, docking studies with CYP51 revealed an interaction between the imidazole ring of the molecules with the heme group as well as the insertion of the chlorinated ring into a hydrophobic cavity at the binding site. These findings suggest that the enzyme, lanosterol 14-demethylase (CYP51), is a potential target for these molecules [107]. An eugenol-based hybrid molecule (**37**, in Figure 5) fused with glycoconjugates developed by Goswami et al. inhibited the conidial and mycelial growth of *A. fumigatus* with a low MIC value of 10.86 µM. Additionally, it was effective against pre-existing fungal biofilms at concentrations between 69.53 and 86.92 µM [108]. Desouza et al. treated eugenol with glycosyl bromide in acetone and lithium hydroxide to synthesize six eugenol derivatives that contained peracetylated glycosides. Among these compounds, the peracetyl glycoside (derivative **38**, in Figure 5) had IC_50_ values that were significantly lower than that of the prototype eugenol and inhibited the development of *Candida albicans*, *Candida glabrata*, and *Candida tropicalis*. Compound **38** demonstrated low cytotoxicity and a selectivity index of 45 against *C. glabrata*, and was found to be 3.4 and 160.0 times more effective than fluconazole and eugenol, respectively [109]. Eighteen novel glucosyl-1,2,3-triazoles hybrid molecules were synthesized by de Magalhes et al. from eugenol and dihydro-eugenol. Their anti-*Candida* spp. activity was assessed using the microdilution method. The eugenol-based triazole **38** in Figure 5 was four times more potent than fluconazole against *C. krusei* and active against *C. glabrata*, *C. tropicalis*, and *C. krusei* at 26.1–52.1 µM. Dihydroeugenol-based derivative **39** in Figure 5 was four times more effective against *C. tropicalis* and *C. krusei* than compound **40**. It displayed a wider range of biological activities than compound **39**. Overall, the molecular modelling studies revealed that compounds **17** antifungal activity is attributed to the inhibition of CYP51, which prevents the formation of fungal ergosterol [110].

### 2.3. Cinnamaldehyde

Cinnamaldehyde (**41**, Figure 6) is a main component of cinnamon oil with antifungal activity [111,112,113,114]. Hybrid molecules containing cinnamaldehyde and other natural compounds have been reported. A recent study revealed cinnamaldehyde’s inhibition effect on *C. albicans* with a MIC value of 0.5 mg/mL. When added to yeast cells at a concentration of 0.031 mg/mL, cinnamaldehyde was as effective as farnesol at preventing dimorphic transformation [115]. 

The existing experimental data indicate that the antimicrobial properties of cinnamaldehyde are attributed to its capability to suppress certain enzyme activities, membrane function, and cell wall biosynthesis [116,117,118,119,120]. A recent study revealed the fourth line of evidence, which indicates that cinnamaldehyde disrupted calcium [Ca^2+^] homeostasis and was implicated in *P. capsici* growth suppression [121]. Experimental data demonstrating the disturbance of intracellular calcium concentration homeostasis revealed altered calcium concentration that inhibited fungal growth [122,123,124]. Numerous investigations have revealed cinnamaldehyde’s interactions with microbial cell membranes, although it is still unclear how this molecule disturbs these membranes. Cinnamaldehyde can also change the lipid composition of microbial cell membranes [125]. *Staphylococcus aureus* cells treated with cinnamaldehyde revealed changes in the membrane lipid profile, increased permeability, and a breakdown of the cell envelope [119]. Other recent research has shown cinnamaldehyde effects on susceptible and resistant fungal strains [126,127]. Microscopic analysis of cells treated with cinnamaldehyde revealed changes in the cellular shape and damage to the plasma membrane and cell wall, providing insight into its mechanism of action [127]. *C. albicans* cells treated with cinnamaldehyde exhibited a modified ergosterol profile [128,129]. Cinnamaldehyde has also been reported to reduce the amount of ergosterol in cell walls in a dose-dependent manner and was effective against a number of fluconazole-resistant clinical isolates [130]. Cinnamaldehyde-based derivatives or hybrid molecules with promising antifungal properties against a wide range of fungal species have been reported by some researchers.

Wani et al. designed and developed a range of azole-based acetohydrazide hybrid compounds containing cinnamaldehyde and assessed their antifungal efficacy. The in vitro evaluation against the resistant clinical isolates of *C. albicans* demonstrated a remarkable antifungal activity of the hybrid molecules (**42**–**48**, Figure 7). Their mechanisms of action proved that these compounds induced apoptosis in *C. albicans* [131]. The effect of compound **44** on the viability and physiology of cell death against *C. auris* was also investigated together with its impact on the cell cycle, oxidative stress enzymes, and the transcriptional profile of the genes that encode these enzymes. The outcomes showed that compound **44** caused fungal cell death with a minimum inhibitory dose of 0.98 μg/mL. Additionally, at sub-inhibitory and inhibitory concentrations, compound **44** reduced the expression and activity of antioxidant enzymes that produced reactive oxygen species and inhibited the cell cycle in *C. auris* at the S and G2/M phases [132].

### 2.4. Curcumin

Curcumin (**49**, Figure 8) is a naturally occurring lipophilic polyphenol compound that has considerable pharmacological effects both in vitro and in vivo via a variety of modes of action. Studies on the antifungal mechanisms of curcumin demonstrated that it inhibits hyphae development by targeting thymidine uptake 1 and induces oxidative stress [42]. It also alters the properties of the membrane-associated enzymes ATPase activity, ergosterol biosynthesis, and proteinase secretion [43]. The pharmacokinetic, pharmacodynamic, and clinical features of curcumin have been identified and described in numerous research studies [133]. Numerous studies have shown that curcumin’s wide range of pharmacological properties includes antifungal [134], anti-inflammatory [135,136], anti-bacterial [137,138], anti-viral [139,140], and anticancer [141] activities.

Curcumin possesses potent antifungal properties against pathogenic fungal strains of *C. albicans*, *Cryptococcus neoformans, Aspergillus* spp., and *Paracoccidioides brasiliensis* [139,140,142,143]. However, issues such as poor acid tolerance and water solubility, low bioavailability, and enzymatic degradation have prevented its development into a clinically effective antimicrobial agent [144]. Several researchers have developed synthetic curcumin derivatives with improved therapeutic benefits by structurally altering the parent curcumin skeleton to address these drawbacks. An outstanding overview of the structures and pharmacological properties of both curcumin analogues was just published by Noureddin et al. The enhanced antitumor, anti-inflammatory, and antioxidant effects of curcumin analogues and hybrids were reported [145]. 

Esmaeelzadeh et al. developed novel hybrid molecules of curcumin attached to the 1,2,3-triazole ring by reacting curcumin with aromatic aldehydes via a Knoevenagel reaction. In comparison to the parent molecule, curcumin, many of the synthesized molecules displayed superior antibacterial and antifungal properties. Among these molecules, compound **63** in Figure 9 demonstrated superior antifungal activity compared to curcumin (173.73 µM), with a MIC of 125.36 µM [146]. Curcumin-based hybrid molecules, **51**–**55** in Figure 9 synthesized by Lal et al. with a significant biological activity were tested against bacterial and fungal strains. These hybrid compounds showed increased cytotoxicity than curcumin [147]. Nagargoje et al. synthesized a library of 2-chloroquinoline-based-monocarbony hybrid molecules of curcumin and tested their in vitro antifungal and antioxidant properties. Most of these molecules demonstrated promising antifungal activity when compared to the widely used antifungal drug, Miconazol. Compounds **56** and **57** in Figure 9 were the most effective. The SAR study showed that their antifungal activity was 2–3 times higher than piperidone and N-methylpiperidone [148]. The hybrid compounds **58**–**61** in Figure 9 developed by Ahmed et al. were 2–4 times more effective against *F. oxysporum*, with MIC values ranging between 15.63 and 31.25 µg/mL for nystatin and ketoconazole [149]. Deshmukh et al. synthesized a variety of dimeric 1,2,3-triazoles with monocarbonyl curcumin hybrid molecules and tested them against the corresponding strains for their in vitro antifungal, antioxidant, and anti-tubercular effects. Only compound **62** in Figure 9 showed promising antifungal efficacy against *Aspergillus niger* with a MIC value of 8 µg/mL, while the majority of compounds showed good antitubercular and antioxidant activity [150].

## 3. Conclusions

Natural products are at the forefront of drug discovery and development due to their wide range of pharmacological activities, particularly their anticancer, antioxidant, antifungal, anti-inflammatory, and antibacterial activities. In terms of antifungal activity, natural products’ effects on fungal strains are via inducing apoptosis, ROS production, mitochondrial dysfunction, and membrane rupture. Natural products are also effective against fungal strains by weakening their cell walls, blocking the production of ergosterol, and interfering with the membrane integrity. However, researchers have not fully explored the potential of natural products due to limitations, such as their poor bioavailability and solubility. In recent years, there has been significant growth in the number of studies on natural products, and several attempts have been made on their structural modifications to enhance their overall biological activities.

This review has discussed the antifungal activities of selected natural products and their synthesized hybrid molecules. The antifungal activities of the discussed natural products were significant when compared to the parent phytochemicals or the reference drug used as a control. Most of the reported natural product-based hybrid molecules displayed synergistic effects when combined with synthetic antifungal drugs or improved antifungal activity when evaluated in vitro against different fungal strains. The reported findings have provided researchers with a good understanding of the various approaches that have been explored for drug development and paved the way for more logical modifications that can be investigated for the development of effective antifungal agents. The exceptional antimicrobial potencies of phytochemicals and their hybrid compounds suggest that they are good candidates for the development of new and effective antifungal drugs. To understand the effectiveness of phytochemicals and their derivatives and pinpoint their molecular targets, more information from in vivo and human studies is needed.

## Figures and Tables

**Figure 1 pharmaceutics-15-02673-f001:**
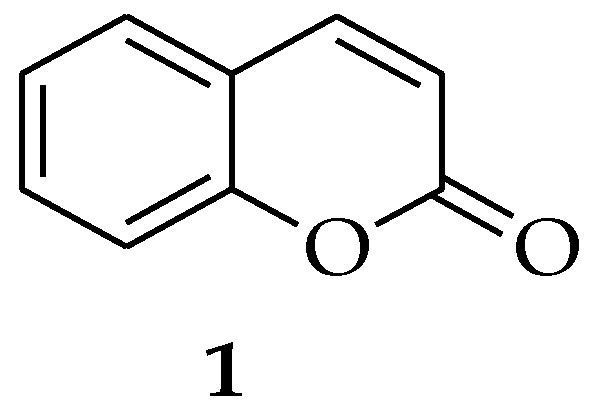
Coumarin chemical structure.

**Figure 2 pharmaceutics-15-02673-f002:**
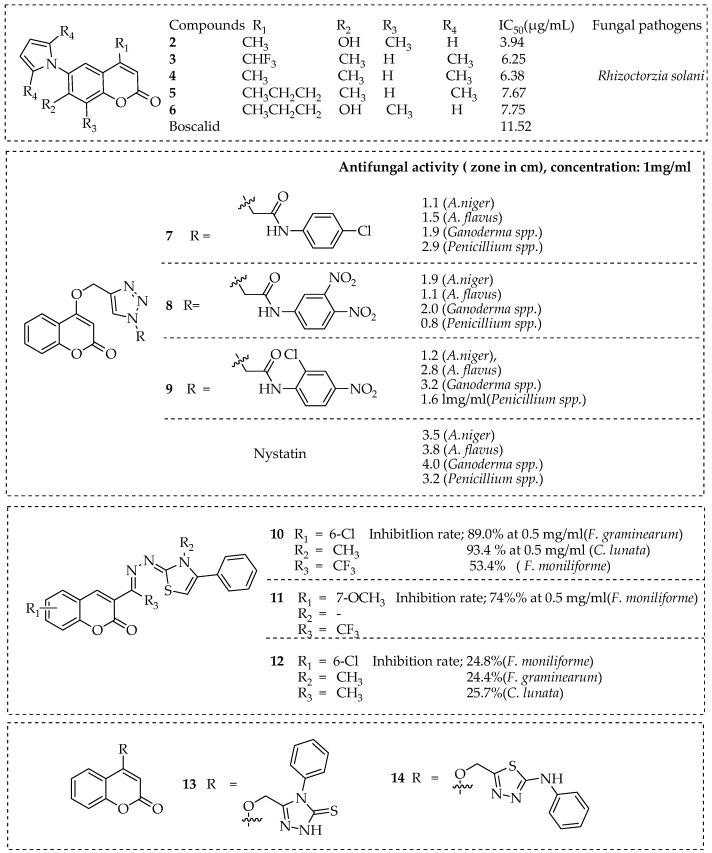
Coumarin-based hybrid molecules with potential antifungal activities.

**Figure 3 pharmaceutics-15-02673-f003:**
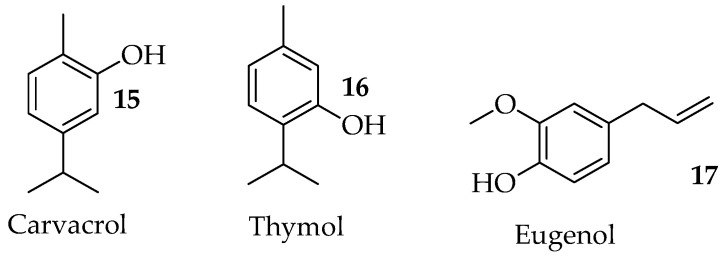
Chemical structure of carvacrol, thymol, and eugenol.

**Figure 4 pharmaceutics-15-02673-f004:**
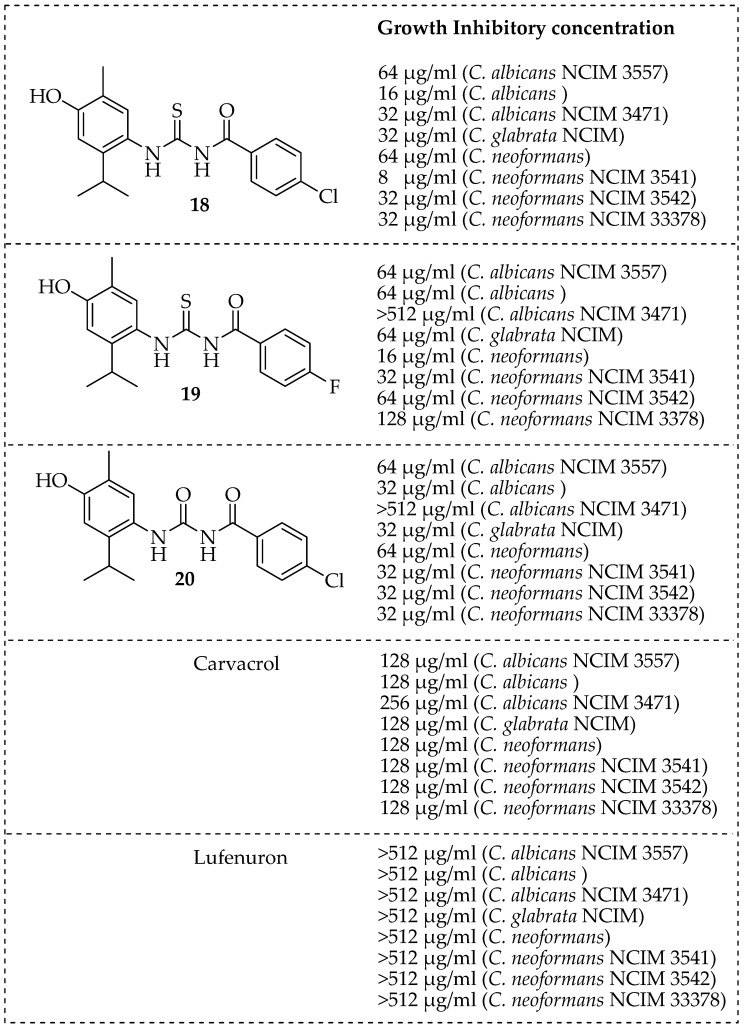
Carvacrol-based hybrid molecules and their antifungal activities.

**Figure 5 pharmaceutics-15-02673-f005:**
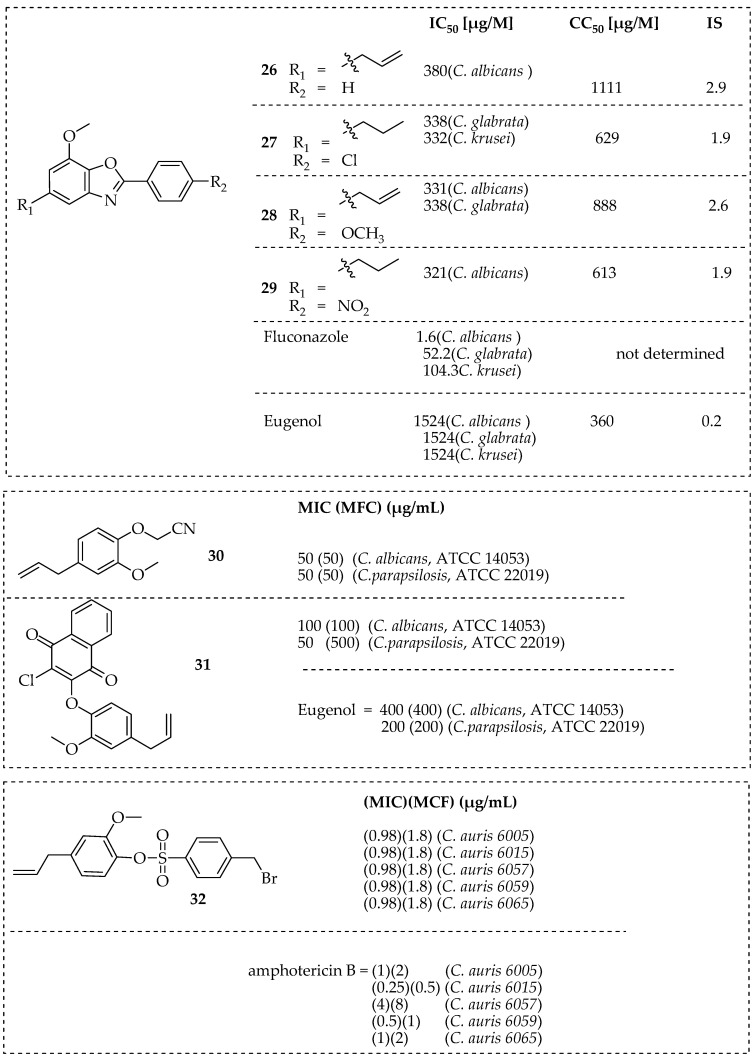
Eugenol-based hybrid molecules and their antifungal activities.

**Figure 6 pharmaceutics-15-02673-f006:**
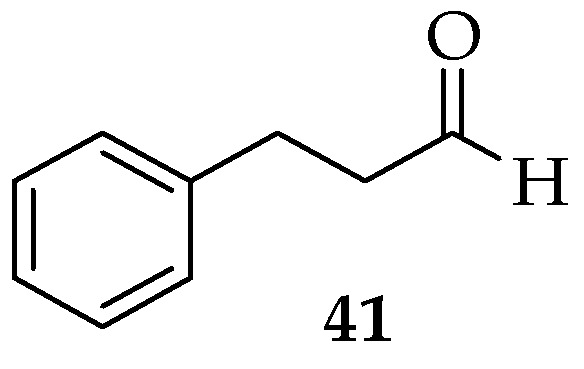
The chemical structure of Cinnamaldehyde.

**Figure 7 pharmaceutics-15-02673-f007:**
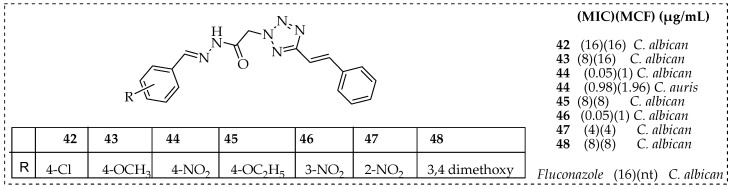
Cinnamaldehyde-based hybrid molecules and their antifungal activities.

**Figure 8 pharmaceutics-15-02673-f008:**
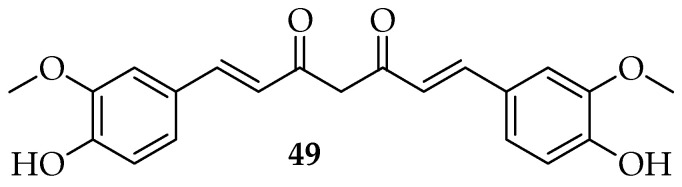
Curcumin’s chemical structure.

**Figure 9 pharmaceutics-15-02673-f009:**
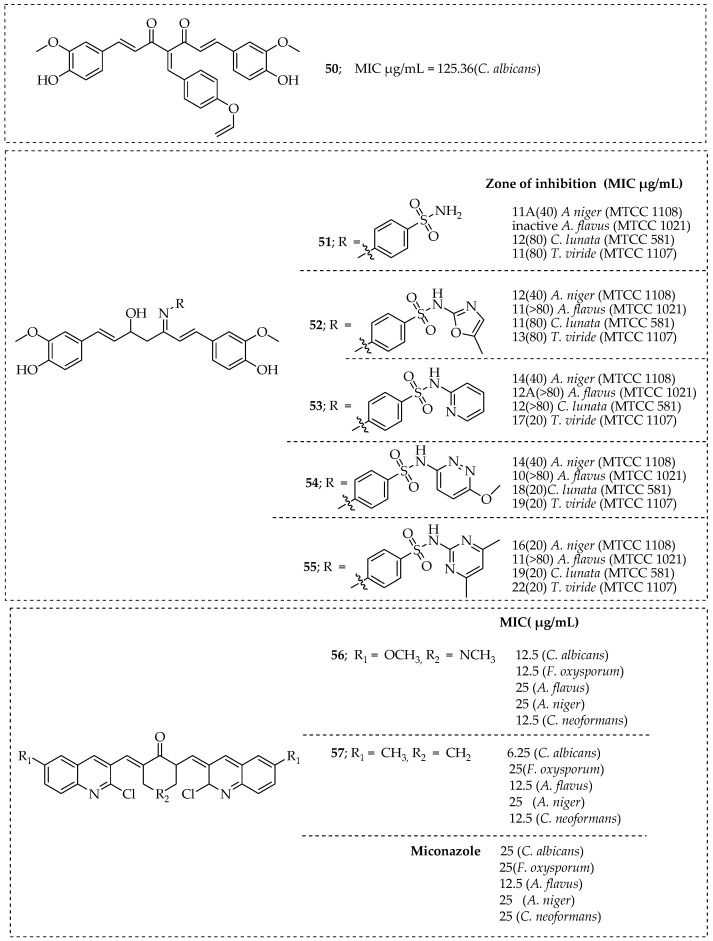
Curcumin-based hybrid molecules and their antifungal activities.

## Data Availability

The data can be shared up on request.

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
