# Peer review of "Antifungal Activities of Natural Products and Their Hybrid Molecules"

_pharmaceutics, 2023, doi:10.3390/pharmaceutics15122673_

Round 1

Reviewer 1 Report

Comments and Suggestions for Authors

Review: Pharmaceutics-2711949

In this work, the authors have reported the potential antifungal activities of hybrid compounds containing natural products based on published literature. This manuscript is important and can be published. However, publication of this manuscript in its present form is not recommended. Some specific points are noted below:

1) In addition to the texts, the compounds described in this paper, can be tabulated in the form of a table.

2) In the conclusion section the authors should briefly mention the new antifungal drugs with unique target mechanisms. Additionally the authors should address the limitation of the natural products and their hybrid molecules.

Minor Points

1) Why Coumarin is used for heading 2. “Natural Products” or something similar can be used for heading 2 and Coumarin should be under subheading 2.1.

Comments on the Quality of English Language

English Language is ok

Author Response

Dear Reviewer 1

Thank you very much for taking the time to review this manuscript. Please find the detailed responses below and the revisions/corrections are highlighted in red font in the re-submitted manuscript.

Comments 1: In addition to the texts, the compounds described in this paper, can be tabulated in the form of a table.

Response 1: Thank you for your suggestion, the selected compounds are listed in a table format as suggested and in the same table, we have also incorporated the antifungal mode of action of these compounds against various fungal pathogens (see introduction).

Comments 2: In the conclusion section the authors should briefly mention the new antifungal drugs with unique target mechanisms. Additionally, the authors should address the limitations of the natural products and their hybrid molecules.

Response 2

Thank you for your kind suggestion, The reported hybrid molecules did not exhibit any distinct target mechanisms but they only demonstrated improved MIC, MCF, and inhibition zone than their parent compounds.

The known limitations of these natural products are their poor bioavailability and solubility which we have mentioned in the conclusion section. Most researchers have designed and developed natural product-based hybrid molecules to overcome these limitations while enhancing their therapeutic effects. However, the data reporting the bioavailability and solubility of the selected natural products and their hybrid molecules is limited. Therefore, No changes have been made in the conclusion.

Comments 3: Why Coumarin is used for heading 2. “Natural Products” or something similar can be used for heading 2 and Coumarin should be under subheading 2.1.

Response 3: Author’s response; we appreciate this suggestion and It has been revised accordingly

Reviewer 2 Report

Comments and Suggestions for Authors

The review article provides an overview on natural compounds used for antifungal purposes, as well as their hybrid forms. While the review article is interesting, some comments need to be addressed:

1- Since the authors did not specify Candida in the title, the introduction section needs also to highlight other common fungal species coming after Candida in prevalence. 

2- The authors referred to resistance to antifungal agents in line 10 of the introduction, however no detailed mechanisms for how this resistance emerges were provided. Please provide an overview of these mechanisms in the introduction section as this is very important for the readers.

3- Figure 2, for compounds 7,8,9, the authors have already written in the title that the concentration is 1 mg/ml, no need to repeat that beside every derivative as this is very confusing.

4- Figure 4 please correct the spelling of fluconazole

5- Figure 5, please ensure that all parts of the figures have the same dimensions. 

6- I am not sure why the authors only selected carvacrol, thymol and eugenol from the terpenoids to talk about, when for example limonene and menthol have reported antifungal activity. Since this is a review article, at least provide a brief summary of the antifungal activities of the compounds mentioned in figure 3.

7- Editing of English language is required

Comments on the Quality of English Language

Moderate editing of English language is required.

Author Response

Dear Reviewer 2

Thank you very much for taking the time to review this manuscript. Please find the detailed responses below and the revisions/corrections are highlighted in red font in the re-submitted manuscript.

Comments 1: Since the authors did not specify Candida in the title, the introduction section needs also to highlight other common fungal species coming after Candida in prevalence. 

Response 1: Authors thank the Reviewer for the comment, we have now included other fungal species in the introduction as suggested.

Comments 2: The authors referred to resistance to antifungal agents in line 10 of the introduction, however no detailed mechanisms for how this resistance emerges were provided. Please provide an overview of these mechanisms in the introduction section as this is very important for the readers.

Response 2: thank you for the kind suggestion, A brief overview of the mechanism of drug resistance has been added in the introduction.

Comments 3: In Figure 2, for compounds 7,8,9, the authors have already written in the title that the concentration is 1 mg/ml, no need to repeat that besides every derivative as this is very confusing.

Response 3: Thank you for the comment, the comment has been revised accordingly

Comments 4: Figure 4 Please correct the spelling of fluconazole

Response 4: the spelling has been corrected, thank you.

Comments 5: In Figure 5, please ensure that all parts of the figures have the same dimensions. 

Response 5 Thank you for the comment, the dimensions of all the figures have been corrected

Comments 6: I am not sure why the authors only selected carvacrol, thymol and eugenol from the terpenoids to talk about, when for example limonene and menthol have reported antifungal activity. Since this is a review article, at least provide a brief summary of the antifungal activities of the compounds mentioned in figure 3.

Response 6 we appreciate the kind suggestion. There are more than 30,000 terpenoids that exist in nature and most of them have reported antifungal activities. Among these terpenoids carvacrol, thymol and eugenol are the most studied terpenoids for antifungal activities which is why we have selected the three. To save readers from being confused, we therefore decided to remove the other terpenoids in Figure 3 and keep only eugenol, carvacrol, and thymol.  Additionally, we have included some information above Figure 3.

Comment 7: Editing of the English language is required
Response 7; thank you for the suggestion, English has been revised as suggested.

Reviewer 3 Report

Comments and Suggestions for Authors

Khwaza and Aderibigbe submitted the manuscript "Antifungal Activities of Natural Products and their Hybrid Molecules," which compiled different classes of natural product compounds, especially coumarins, essential oil, cinnamaldehyde, and curcumins, and their hybrid structures. They demonstrated an excellent chemical representation of such hybrid antifungal structures with their antifungal activities. 

There are a few minor comments for the authors.

1. In most figures, authors often use growth inhibition or zone of inhibition values (which sometimes have arrows), making it complex to understand. Could you please present these values in an easy-to-understand manner, as they are a bit confusing in their current form?

2. One point missing from the manuscript is the mode of action of these antifungals. Please provide a brief description if those are known. 

3. There are some typographical errors in the figures. For example, some of the MIC values have different writing styles, and possibly cross-check the structures from the references (to make sure if they are correctly redrawn).

4. Generalised information on human fungal disease should be included before starting the main topics of the manuscript. As I see, some of the fungi have a higher infection rate. At the same time, some of them do not pose any threat. Therefore, it would improve understanding among readers before going to the main section of the manuscript if the authors agree to include some information on pathogenic fungi (about human infection), which would improve the readability of the paper.

Author Response

Dear Reviewer 3

Thank you very much for taking the time to review this manuscript. Please find the detailed responses below and the revisions/corrections are highlighted in red font in the re-submitted manuscript.

Comment 1: In most figures, authors often use growth inhibition or zone of inhibition values (which sometimes have arrows), making it complex to understand. Could you please present these values in an easy-to-understand manner, as they are a bit confusing in their current form?

Response 1; thank you for the suggestion, The arrows in all the figures have been removed.

Comment 2: One point missing from the manuscript is the mode of action of these antifungals. Please provide a brief description if those are known. 

Response 2; although the mechanism of action of these antifungals has not been thoroughly researched, we have tried to give the mechanism of action in a table format under the introduction.

Comment 3: There are some typographical errors in the figures. For example, some of the MIC values have different writing styles and possibly cross-check the structures from the references (to make sure if they are correctly redrawn).

Response 3; thank you for the suggestion, All the MIC values in the figures have been corrected

Comment 4: Generalised information on human fungal disease should be included before starting the main topics of the manuscript. As I see, some of the fungi have a higher infection rate. At the same time, some of them do not pose any threat. Therefore, it would improve understanding among readers before going to the main section of the manuscript if the authors agree to include some information on pathogenic fungi (about human infection), which would improve the readability of the paper.

Response 5; thank you for the kind suggestion the information on human fungal deceases has been added in the introduction.